# Research into a Multi-Variate Surveillance Data Fusion Processing Algorithm

**DOI:** 10.3390/s19224975

**Published:** 2019-11-15

**Authors:** Yi Mao, Yi Yang, Yuxin Hu

**Affiliations:** State Key Laboratory of Air Traffic Management System and Technology, Nanjing 210000, China; mao_y@nuaa.edu.cn (Y.M.); helenhu_yuxin@163.com (Y.H.)

**Keywords:** ADS, radar, data integration, wave filtering

## Abstract

Targeted information sources include radar and ADS (Automatic Dependent Surveillance) for civil ATM (Air Traffic Management) systems, and the new navigation system based on satellites has the capability of global coverage. In order to solve the surveillance problem in mid-and-high altitude airspace and approaching airspace, this paper proposes a filter-based covariance matrix weighting method, measurement variance weighting method, and measurement-first weighted fusion method weighting integration algorithm to improve the efficiency of data integration calculation under fixed accuracy. Besides this, this paper focuses on the technology of the integration of a multi-radar surveillance system and automated related surveillance system in the ATM system and analyzes the constructional method of a multigeneration surveillance data integration system, as well as establishing the targeted model of sensors and the target track and designing the logical structure of multi-radar and ADS data integration.

## 1. Introduction

A new flight system is composed of four parts: communications, navigation, ADS and ATM. ADS [1] (Automatic Dependent Surveillance) measures the airplane’s 4D positions through the navigation and positioning system, sends relevant data automatically to the ground ATM center via the air communication data chain, and then realizes ATM and flow management. For airplanes without ADS and those for military purposes, radar is still the major approach to obtain information. Therefore, after the application of the new flight system, there are two surveillance methods: radar and ADS. ADS and multi-radar data fusion comprise the fundamental and key technologies in the system. 

In recent years, the KF (Kalman Filter) [2], MKF (Mixture Kalman Filter) [3], EKF (Extended Kalman Filter) [4,5] have been used for multi data fusion. Ma and Wang proposed an RFID (Radio Frequency Identification) tracking method which combines the received signal strength with phase shift to predict the instantaneous position of a moving target. It is assumed that the trajectory of the target can be approximated by a series of lines, and the instantaneous velocity of the target can be estimated by fusing the rough position; furthermore, the Kalman Filter is used to improve the accuracy of position estimation. As an effective filtering technique, MKF is usually used for state estimation in CODLS (conditional dynamic linear systems). Yu proposed an improved distributed MKF for state estimation in CODLS. Because global likelihood is not suitable for distributed computing in CODLSs, the volume criterion is used to calculate likelihood in order to realize distributed likelihood computing. EKF is a nonlinear filtering method based on Taylor expansion approximation, which is widely used in nonlinear systems. Xing and Xia used theoretical analysis and simulation experiments to compare the data fusion performance of EKF and calibrated an unscented Kalman filter by calculating the trace of error covariance and absolute mean square error.

Nonlinear filtering algorithms based on Monte Carlo simulation, such as Particle Filters (PF) [6,7,8] and Box Particle Filters (BPF) [9,10,11], have also been used in multi-sensor target tracking and data fusion. The PF algorithm can approximate the posterior probability density function of a state by extracting random particles. It can deal with the state estimation of a non-Gaussian system without the assumption that the non-linear system satisfies Gaussian distribution. Li used sequential importance sampling PF to cross-process the number of visible optical communications based on positioning and inertial navigation to obtain accurate target location information. Experts and scholars collaborate the sequential Monte Carlo method and use interval analysis to propose BPF; this non-linear filtering method has a good effect when dealing with an inaccurate random measurement system. Compared with standard PF, BPF has the great advantage of reducing computational complexity and being more suitable for distributed filtering systems. In some application scenarios, PF requires thousands of particles to achieve considerable accuracy and stability, while KPF only needs dozens of box particles to achieve the same accuracy. Gning uses Bernoulli PF and Bernoulli KFP for target detection and tracking. The simulation results show that the posterior probability density function of the target state can be accurately estimated by using the two fusion algorithms, although KFP has higher fusion efficiency than PF. Zhang uses BPF to deal with multi-target tracking under strong clutter surveillance under the assumption of non-linearity.

For mid-and-high altitude airspace and approaching airspace, some planes are only equipped with a radar transponder, while some are only equipped with radar airborne equipment; besides this, some ATC radars are unable to cover all airspace heights, and thus an ADS ground station is required for monitoring and compensating. Thus, this paper considers the fusion algorithm for the signals of ATC radar and ADS-B in order to reduce the observation error and improve the global monitoring capability.

The data fusion of multi-radar and ADS must take into consideration the relationship between each sensor’s and system’s coordinates, forming a synchronism of different sensors obtaining the same target information, the unification of measurement units, connection, relevance, filtering and the evaluation of multi-sensors and multi-targets. This paper proposes a filter-based covariance matrix weighting method, measurement variance weighting method, and measurement-first weighted fusion method weighting integration algorithm that can remarkably reduce the errors of observed tracks and the ensure precision of observation systems. It is known that the working mode of civil ATC radar can divided into modes A, S, and C: mode A is used for ground broadcast inquiry, mode S is used for selective addressing inquiry, and mode C is only used for height inquiry while modes A and S are working. All the information achieved by the ground station amounts to the normalized quadratic code set after pre-processing; thus, this paper will not consider the effect of different modes.

## 2. Relationship between Sensor Coordinates and System Coordinates

The Globe is an uneven ellipsoid which can be approximately expressed as a WGS84 ellipsoid for precise calculation. Radar stations and ATM centers located at different longitudes and latitudes are scattered points on the ellipsoid’s surface that have corresponding tangent planes. The targets measured by the radar station are expressed in the radar-centered coordinates. The target’s position is marked with distance, orientation, elevation or height (*ρ*,*α*,*θ*), as shown in Figure 1.

ADS reports the target’s information in the form of longitude, latitude and height (*λ*,*φ*,*H*), while the ATM center has independent system coordinates (*X*,*Y*,*Z*). First, the ATM center unifies the same target’s coordinates reported by different sensors in space; i.e., it transforms the sensor’s target to the ATM center’s system coordinates.

Considering the small deviation between the ground tangent plane of each sensor and the tangent plane of the ATM center when ATM covers a small range, the sensor’s target (*X_T_*,*Y_T_*) can be approximately expressed as (*X_S_*,*Y_S_*) in system coordinates, where $X_{S}=X_{T}+\Delta X$, $Y_{S}=Y_{T}+\Delta Y$, (∆*X*,∆*Y*) is the position difference of the origin in the coordinates.

Set the ATM center’s geographical longitude and latitude as (*λ*_0_,*Φ*_0_), and the radius of latitude as $\gamma_{0}$; set the sensor’s geographical longitude and latitude as (*λ*_1_,*Φ*_1_ ) and the radius of latitude as $\gamma_{1}$. Then,
(1)$$\begin{array}{l} \Delta X=\frac{1}{2}[\gamma_{0}(\lambda_{1}-\lambda_{0})+\gamma_{1}(\lambda_{1}-\lambda_{0})]\\ \approx \frac{a^{2}}{2}[\frac{1}{\sqrt{a^{2}+b^{2}atg^{2}\phi_{0}}}+\frac{1}{\sqrt{a^{2}+b^{2}atg^{2}\phi_{1}}}]\Delta \lambda \cdot \frac{\pi }{180}\\ \approx \frac{1}{2}(\gamma_{0}+\gamma_{1})\Delta \lambda \cdot \frac{\pi }{180} \end{array}$$
(2)$$\Delta Y=(\Phi_{1}-\Phi_{0})\cdot 60\cdot 1852(m)$$
where (*λ*,*Φ*) adopt the degree as the unit, accurate to 0.0001 degree. ∆*X*, ∆*Y* adopt meters as the unit: *a* = 6,378,137 ± 2 m; *b* = 6,356,752.31 m.

## 3. Structural Design of Data Fusion and System Modeling

### 3.1. Design of Logic Structure of Data Fusion

The classification of a single sensor’s flight tracks is locally done by distributed sensors in this system. Seen as Figure 2, after the state of the single sensor’s target, such as airport surveillance radar(ASR) or secondary surveillance radar(SSR), enters the system via a system communication processor, it enters the front processing module first. The document of the target’s data based on the uniform time and space is then created. After track correlation and fusion processing, the estimation of the target’s position is approximate to the actual position. After the fusion of each sensor’s target characteristics and transcendental planning information, false information is eliminated and true information is retained. Characteristic information is shared to a maximal extent; then, the fusion of the target’s geometric position and characteristics level is completed, as well as the estimation of the target’s state. The determination of the temporal and spatial status of the airplane within the ATM zone is then complete and clear. 

The target’s state value is correlated to the flight plan. The airplane’s current position can be estimated, and the consistency, conflicts, or potential conflicts between the airplane and flight plan can be detected according to certain rules. Then, corresponding solutions to the conflicts are offered to help with the controller’s decision-making, as well as the complete decision-level fusion of the target data.

### 3.2. Zone Division of Data Fusion Processing

In order to ensure that any target within the ATM zone is processed with equal probabilities, three categories of targets are updated according to certain levels and rules. The ATM zone is divided into multiple sub-zones that are processed in order. 

The radar makes a circular scan of the polar coordinate system. Data of the measured target are obtained in the order of distance (near to far) and orientation (gradual increasing or decreasing). Thus, in single-radar data processing, in order to ensure the instantaneity of data processing, the 360° plane is generally divided into multiple sections. Synchronizing to the radar scan, each section’s target data is processed successively. In terms of the principle of section division, on the one hand, the section’s angle should be small enough to ensure a limited number of targets within each section; on the other hand, considering the difference between the estimated position and observed position, the section’s angle should be large enough. In general, 360° is divided into 32 sections, of 11.25° each. After the scanning of each section, the previous two to three sections’ target tracks are processed hierarchically. 

In contrast to the single-radar zone, there are a number of radars in the ATM zone that carry out circular non-synchronous and non-homocentric circular scanning. Airplanes’ ADS target reports on different tracks are independent. The time of the target’s arrival at the system is not sequenced in the increasing order of orientation; hence, it is obviously improper to process tracks by dividing the zone into sub-sections based on the single-radar’s polar coordinate system scanning, which is not beneficial for realizing the instantaneity and agility of processing. 

According to the above principle of the division of the single-radar zone, the ATM zone is divided into multiple rectangular bars in the orientation of increasing latitude. Every bar’s width is 1° in latitude, as shown in Figure 3. 

Regarding the time of the system’s target fusion processing (the time of one target’s processing in the entire zone), the minimal time of radar scanning time in the system (Ti) is chosen: Ti = Tmin. The rate of the target data updating in the system is Tmin. The advantages lie in the small amount of system delay and the avoidance of processing a single-radar’s repeated target report.

### 3.3. Target Model of Sensors and Target Track

In the target’s motion model, the variables of state are the target’s position and speed. The accelerated speed can be regarded as a disturbance input with random characteristics that are attributed to the non-predictability of the driver and surrounding environment disturbance. Suppose that the accelerated speed at one moment is irrelevant to that at other moments and that it is a Gaussian distribution stationary random process with zero mean and variance *σ_a_*^2^; i.e.,
(3)$$\begin{array}{c} E\left\{a_{x,k}\right\}=0\\ E\left\{{\left(a_{x,k}\right)}^{2}\right\}=\sigma_{a}^{2}\\ E\left(a_{x,k},a_{x,n}\right)=0,n\neq k \end{array}$$

The target’s motion model is
(4)$$\begin{array}{l} S_{k+1}=\Phi S_{k}+G_{k}U_{k}\\ \Phi =\left[\begin{array}{cccc} 1 & 1 & 0 & 0\\ 0 & 1 & 0 & 0\\ 0 & 0 & 1 & T\\ 0 & 0 & 0 & 1 \end{array}\right] \end{array}$$
where *S* is state vector, *S*^T^ = [*X_k_*,*Y_k_*], Φ is transposed matrix, T is the interval of target sampling, *G_k_* is the noise gain matrix, and *U_k_* is the noise process.

The target’s observation model is *Z_k_*_+1_ = *HS_k_*_+1_ + *W_k_*_+1_, where *H* = [1,0] is the projection vector from the state to observation and W is observation noise. Given that *G_k_*, *U_k_* is an unknown quantity with statistical characteristics, it can be ignored when the model is approximate. The ignored item can be compensated by adaptive filtering.

For sensor *i*, Kalman filtering is applied. The target’s position and speed are optimally estimated as follows.

*N* sensors are able to obtain the same goal’s state estimation and co-variance matrix at time *K*; thus, combining partial tracks by making use of relevant information proceeds via the fusion of the tracks.

## 4. Radar Modification of Target Position

Due to the sensors’ different geographical positions, a single sensor’s target position should be transformed as (*λ*,*φ*,*H*), and then (*X_S_*,*Y_S_*,*Z*) in the system, which is done by the system’s front processing module.

Noticeably, before the transformation is completed by system’s front processing module, it is necessary to project the position of the single sensor’s target [*ρ*,*α*,*θ* (*H*)] to obtain (*X_P_*,*Y_P_*,Z), as shown in Figure 4. Only after this can the transformation of system coordinates be carried out. ADS reports the target in the form of (*λ*,*φ*,*H*), which is a standard geographical coordinate that can directly be processed in the system without any modifications.

The target position measured by the radar is the slope distance *ρ* from the target to radar, relative to the azimuthal angle *α* in the north. For the secondary radar, the target’s height is the C-model height relative to sea level, while the primary radar generally gives the target’s elevation angle *θ*. The task of the radar modification of the target position is to project the target’s spatial position to the radar’s plane, in which the stereographic projection method is adopted.

### 4.1. Relationship between Target Height and Radar Antenna Elevation Angle

According to Figure 4, it can be deduced that
(5)$$Z=\rho \sin\theta +a+\frac{\rho^{2}}{2\left(R+a\right)}$$

As shown in Figure 5, *ρ* is the target’s slope distance, *θ* is the target’s elevation angle, *a* is the radar’s height, and *R* is the radius of the globe.

### 4.2. Target’s Planar Projection Modeling

According to Figure 5, $\frac{D}{\rho }=\sqrt{\frac{1-Z^{2}}{\rho^{2}}}$, the target’s position on the radar’s plane is *X* = *ρ*cos*α*, *Y* = *ρ*sin*α*; (*Xq*,*Yq*) is
(6)$$\begin{array}{l} X_{q}=X\sqrt{1-\frac{Z^{2}}{\left(X^{2}+Y^{2}\right)}}=K_{1}X\\ Y_{q}=Y\sqrt{1-\frac{Z^{2}}{\left(X^{2}+Y^{2}\right)}}=K_{1}Y \end{array}$$

### 4.3. Projection of the Target on the Stereographic Plane

The target’s spatial position is projected to the *Q* point on the plane and finally transformed to point *P*(*Xρ*,*Yρ*) with the stereographic projection method. The projection from target’s spatial position to the radar station’s plane is thus completed.

From *d* = *2Rtg*(*β*/*2*), it can be deduced that
(7)$$d=2R\frac{D}{2R+2Z-\frac{\rho^{2}}{2R+2a}}=K_{2}D$$

Meaning that
(8)$$\begin{array}{l} \{\begin{array}{c} X_{\rho }=K_{2}X_{q}=k_{1}k_{2}X\\ Y_{\rho }=K_{2}Y_{q}=k_{1}k_{2}Y\\ Z=\rho \sin\theta +a+\frac{\rho^{2}}{2R+2a} \end{array} \end{array}$$
where $k_{1}=\sqrt{1-\frac{Z^{2}}{\rho^{2}}}$, $k_{2}=R/(R+Z-\frac{\rho^{2}}{4(R+a)})$, X=ρcosa, and Y=ρsina.

## 5. Weighted Fusion Algorithm

As stated above, under the self-contained fusion structural model, each sensor processes the data of the measured target. The processing results of the same target’s data by *N* sensors are then analyzed by the system’s front processing module. After temporal and spatial alignment, the estimation of the target’s state at time *k*, ${\overset{⌢}{X}}_{i}(k|k)$ and covariance matrix $P_{i}(k)(k|k)$, *i* = *1*, *2*, … *N*, (*N* is the number of sensors covering the target) are obtained. The covariance matrix of errors of each sensor’s observation of the target is $P_{i}(k)(k)$, For the purpose of system simplification, $R_{k}(k)$ can be simplified as the standard deviation of the observation value $\sigma W^{2}$. Track fusion is completed by inputting ${\overset{⌢}{X}}_{i}(k|k)$ and $P_{i}(k)$ into the system. Owing to different filtering methods deployed by different sensors, some sensors may have no filtering co-variance matrix P(k|k). Besides, due to the different distances between sensors and the target or different positions, different filtering covariance matrixes $P_{i}(k|k)$ or measurement covariance matrixes $P_{i}(k)$, may be obtained. The covariance matrix presents differences in terms of the precision of sensor track data. To carry out track fusion with $P_{i}^{-1}(k|k)$ or $R_{i}^{-1}(k)$ as weighting factors, we use the weighted fusion algorithm.

### 5.1. Filtering Covariance Matrix Weighted Method

The fusion processing of sensor *i* and sensor *j* is conducted first; ${\overset{⌢}{x}}_{i}(k|k),{\overset{⌢}{x}}_{j}(k|k),P_{i}(k|k),P_{j}(k|k)$ are known. Set $\overset{⌢}{x}(k)$ as the result of fusion: it is necessary to calculate $\overset{⌢}{x}(k)$ and the covariance matrix P(k|k). With the linear minimum mean squared error, it can be estimated that
(9)$$\hat{x}(k)=\bar{x}+PxzP-lzz(Z-\bar{z})$$
where $\bar{x}$ is the transcendental mean value of X(k), and $\bar{z}$ is the transcendental mean value of X(k) measurement. $X_{i}(k|k)$ is regarded as $\bar{x}$, while $X_{j}(k|k)$ is regarded as measurement. It can be deduced that
(10)$$\begin{array}{ll} \hat{x}(k)= & {\hat{x}}_{i}(k|k)+[P_{i}(k|k)-P_{ij}(k|k)]{\left[P_{i}(k|k)+P_{j}(k|k)-P_{ij}(k|k)-P_{ij}^{T}(k|k)\right]}^{-1}\\ & \cdot [{\hat{x}}_{j}(k|k)-{\hat{x}}_{i}(k|k)] \end{array}$$
(11)$$\begin{array}{ll} P(k|k)= & P_{i}(k|k)-[P_{i}(k|k)-P_{ij}(k|k)]{\left[P_{i}(k|k)+P_{j}(k|k)-P_{ij}(k|k)-P_{ij}^{T}(k|k)\right]}^{-1}\\ & \cdot [P_{i}(k|k)-P_{ij}^{T}(k|k)] \end{array}$$
where $P_{ij}(k|k)=P_{ij}^{T}(k|k)$, which is the mutual correlation error covariance of sensor *i*’s and sensor *j*’s partial tracks caused by common system noise. It can be considered that the partial tracks of sensor *i* and sensor *j* are not correlated. The result of fusion is:(12)$$\hat{x}(k)={\hat{x}}_{i}(k|k)+P_{i}(k|k){\left[P_{i}(k|k)+P_{j}(k|k)\right]}^{-1}\cdot [{\hat{x}}_{j}(k|k)-{\hat{x}}_{i}(k|k)]$$
(13)$$P(k)=P_{i}(k|k)-P_{i}(k|k){\left[P_{i}(k|k)+P_{j}(k|k)\right]}^{-1}\cdot P_{i}(k|k)$$

The above method is applied in *N* sensors’ processing; i.e., the filtering covariance matrix weighted method. The result is as below:(14)$$\{\begin{array}{c} \hat{x}(k)=P(k)\sum_{i=1}^{N}P_{i}^{-1}(k|k){\hat{x}}_{i}(k|k)\\ P^{-1}(k)=\sum_{i=1}^{N}P_{i}^{-1}(k|k) \end{array}$$

### 5.2. Measurement Variance Weighted Method

Supposing that there is no noise in the system and that measurement variance is a constant, $E\left[V_{i}(k+1)V_{i}^{T}(k+1)\right]=R_{i}=\mathrm{const}$, Kalman filtering is degraded to the least square method. The target moves in straight line at a constant speed. Under this condition, $P_{i}(k|k)=p(k)R_{i}$. Thus, $p(k)=\left[\begin{array}{cc} \frac{2(2k-1)}{k(k+1)} & \frac{6}{k(k+1)T}\\ \frac{6}{k(k+1)T} & \frac{12}{k(k^{2}-1)T^{2}} \end{array}\right]$, and the results of fusion are
(15)$$P^{-1}(k)=\sum_{i=1}^{N}P_{i}^{-1}(k|k)=\sum_{i=1}^{N}P^{-1}(k)R_{i}^{-1}=P^{-1}(k)\sum_{i=1}^{N}R_{i}^{-1}$$
(16)$$\sum_{i=1}^{N}P_{i}^{-1}(k|k){\hat{x}}_{i}(k|k)=\sum_{i=1}^{N}P^{-1}(k)R_{i}^{-1}{\hat{x}}_{i}(k|k)=P^{-1}(k)\sum_{i=1}^{N}R_{i}^{-1}{\hat{x}}_{i}(k|k)$$

Substituting Equations (15) and (16) into Equation (14), we obtain the following:(17)$$\{\begin{array}{c} \hat{x}(k)={\left(\sum_{i=1}^{N}R_{i}^{-1}\right)}^{-1}\sum_{i=1}^{N}R_{i}^{-1}{\hat{x}}^{-1}(k|k)\\ P^{-1}(k)=p^{-1}(k)\sum_{i=1}^{N}R_{i}^{-1} \end{array}$$

After the measurement variance weighting, the calculation quantity of filtering covariance matrix weighting is greatly simplified, with equivalent results.

### 5.3. Measurement-First Weighted Fusion

The following can be deduced from Equation (17):(18)$$\begin{array}{ll} \hat{x}(k) & ={\left(\sum_{i=1}^{N}R_{i}^{-1}\right)}^{-1}\sum_{i=1}^{N}R_{i}^{-1}{\hat{x}}^{-1}(k|k)\\ & ={\left(\sum_{i=1}^{N}R_{i}^{-1}\right)}^{-1}\sum_{i=1}^{N}R_{i}^{-1}{\hat{x}}^{-1}(k|k-1)\\ & +K(k)[{\left(\sum_{i=1}^{N}R_{i}^{-1}\right)}^{-1}\sum_{i=1}^{N}R_{i}^{-1}Z_{i}(k)-{\left(\sum_{i=1}^{N}R_{i}^{-1}\right)}^{-1}\sum_{i=1}^{N}R_{i}^{-1}H{\hat{x}}^{-1}(k|k-1))] \end{array}$$

Since ${\overset{⌢}{x}}_{i}(k|k-1)=\Phi (k){\overset{⌢}{x}}_{i}(k-1|k-1)$, thus,
(19)$${\left(\sum_{i=1}^{N}R_{i}^{-1}\right)}^{-1}\sum_{i=1}^{N}R_{i}^{-1}{\hat{x}}^{-1}(k|k-1)=\Phi (k){\left(\sum_{i=1}^{N}R_{i}^{-1}\right)}^{-1}\sum_{i=1}^{N}R_{i}^{-1}{\hat{x}}^{-1}(k-1|k-1)$$

Substituting Equation (19) into Equation (18), we obtain
(20)$$\{\begin{array}{c} \hat{x}(k)=\hat{x}(k|k-1)+K(k)[{\left(\sum_{i=1}^{N}R_{i}^{-1}\right)}^{-1}\sum_{i=1}^{N}R_{i}^{-1}Z_{i}(k)-H{\hat{x}}_{i}(k|k-1))]\\ P(k)=p(k){\left(\sum_{i=1}^{N}R_{i}^{-1}\right)}^{-1} \end{array}$$

If $(\sum_{i=1}^{N}R_{i}^{-1}{)}^{-1}\sum_{i=1}^{N}R_{i}^{-1}Z_{i}(k)$ is regarded as total measurement, Equation (11) is equivalent to the general Kalman filtering equation. The method conducts measurement fusion first and then filtering. The initial value is $P(0)=\sum_{i=1}^{N}R_{i}^{-1}P_{i}(0)$, $X(0)=(\sum_{i=1}^{N}R_{i}^{-1}{)}^{-1}\sum_{i=1}^{N}R_{i}^{-1}{\overset{⌢}{x}}_{i}(0)$.

The calculation quantity of the measurement-first weighted fusion algorithm is minimal, with unchanged precision.

## 6. Numerical Simulation Analysis

The paper conducts multi-radar and ADS data fusion processing simulation concerning track samples under various conditions. The results indicate that the weighted fusion algorithm is practical and feasible and that the calculation is remarkably quicker than the current method. 

We import 15 actual tracks of aircrafts, under the surveillance of six radars and ADS in total. The radar positions and actual tracks are shown in Figure 6, in which black tracks are the actual flight tracks of the target. It is supposed that all targets fly in a straight line at a constant speed. Red points are the radars’ positions on the ground. The sampling period of a radar and ADS is set as 1 s. This paper assumes that the observation errors of the radar and ADS are 60 dB and 40 dB, respectively, and both of them are Gaussian white noise. There is no acceleration of the aircraft during the flight phase, and the initial velocity is 300 m/s.

For the purpose of a clear display, only the observed track of the target’s track 1 is given below. Figure 7 shows the tracks observed by radar and ADS on the horizontal and longitudinal plane and actual tracks. Figure 8 shows the observed track calculated with the weighted fusion algorithm. From the figures, it can be evidently seen that the observed track is more approximate to the actual track with the weighted fusion algorithm.

Figure 9 shows the error of the radar observation at each site, the error of ADS observation and the comparison of the errors of observation with weighted fusion. It can be found that the error is reduced obviously by means of weighted fusion.

Figure 10 shows the comparison of the errors of the observed tracks of each target to verify that the weighted fusion algorithm is able to greatly improve the observation longitude in terms of different target tracks.

Figure 11 shows the calculation time by means of multi-radar and ADS fusion, adopting the weighted fusion algorithm and one-by-one radar fusion. It can be seen that the weighted fusion algorithm largely reduces the calculation quantity and shortens the calculation time on the prerequisite of ensuring the precision of observation.

## 7. Conclusions

Multi-sensor data fusion processing is a core technology in different information processing systems, characterized by high requirements of theoretical and technological mastery. The effects are closely related to the overall system design. The paper makes a simulation calculation of multi-radar and ADS data fusion processing in terms of historical track samples under different conditions. The results indicate an evident reduction of errors of observation after optimization by the weighted fusion algorithm. Under the prerequisite of ensuring the precision of observation, the calculation time by using the weighted fusion algorithm is obviously shortened compared to the traditional algorithm. The experimental results indicate the following: (1)The weighted fusion algorithm can remarkably reduce the errors of observed tracks and ensures the precision of observation systems.(2)With equal precision of observation, the calculation quantity of the weighted fusion algorithm is greatly reduced compared to the traditional fusion algorithm, thus effectively improving the system’s overall surveillance ability.

## Figures and Tables

**Figure 1 sensors-19-04975-f001:**
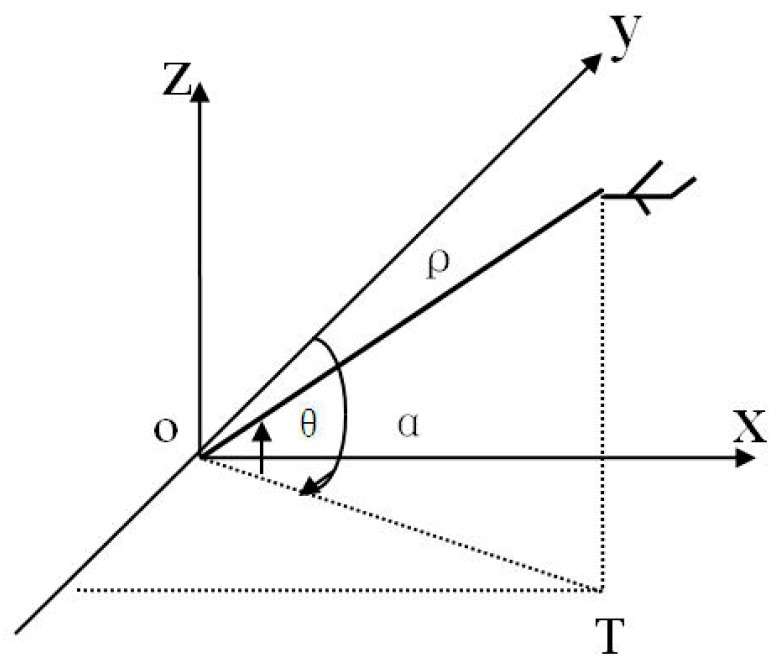
Coordinates of a single radar target.

**Figure 2 sensors-19-04975-f002:**
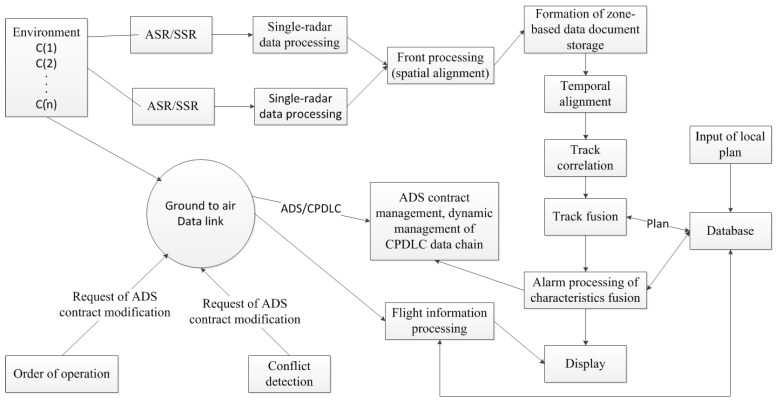
Logic structure of multi-radar and Automatic Dependent Surveillance (ADS) data fusion.

**Figure 3 sensors-19-04975-f003:**
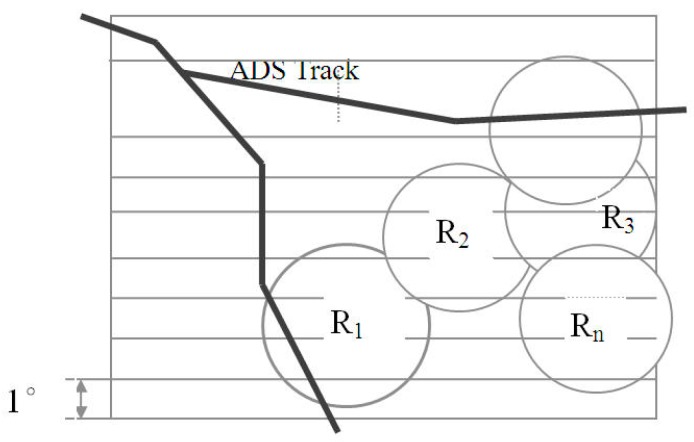
Division of Air Traffic Management (ATM) zone.

**Figure 4 sensors-19-04975-f004:**
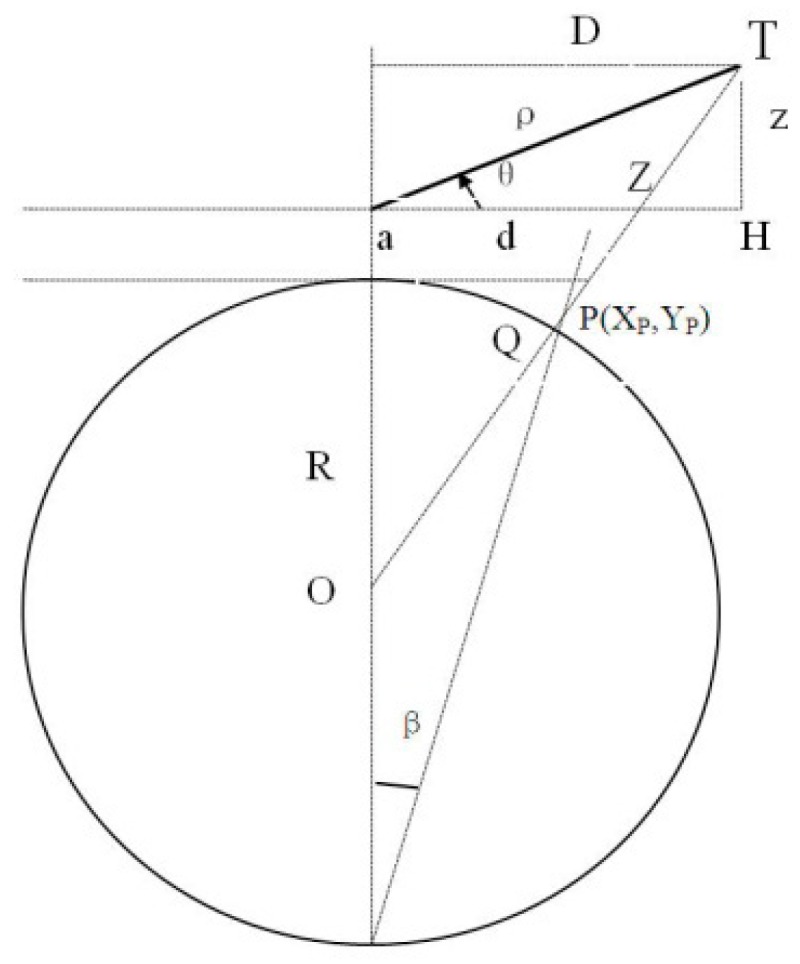
Target projection.

**Figure 5 sensors-19-04975-f005:**
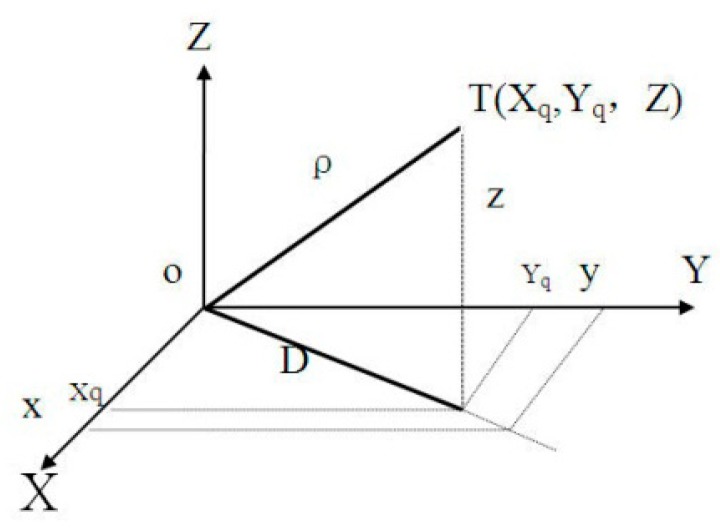
Target’s planar projection.

**Figure 6 sensors-19-04975-f006:**
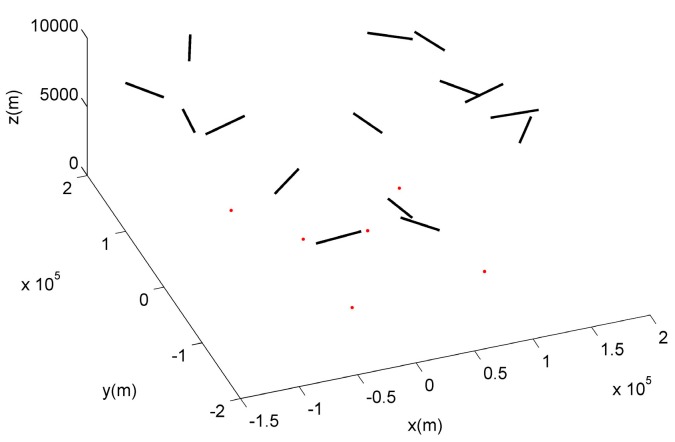
Actual tracks of target and allocation of radars.

**Figure 7 sensors-19-04975-f007:**
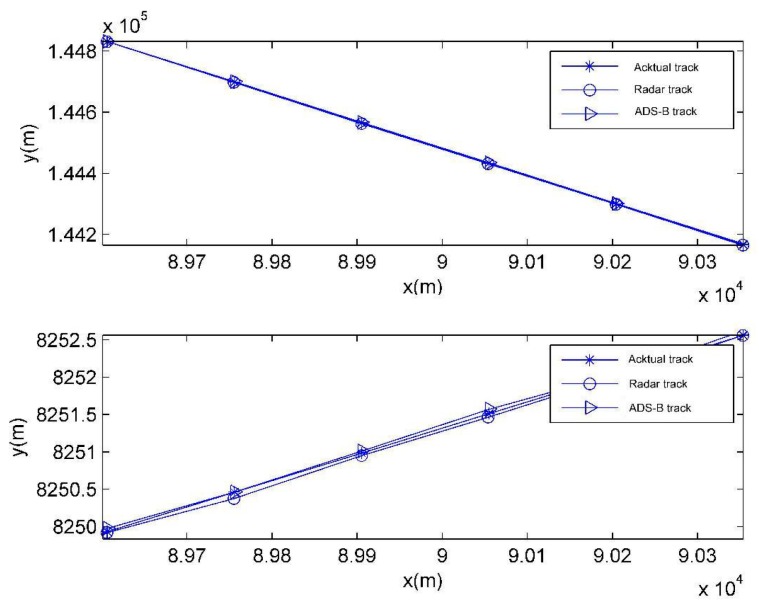
Observed track and actual track.

**Figure 8 sensors-19-04975-f008:**
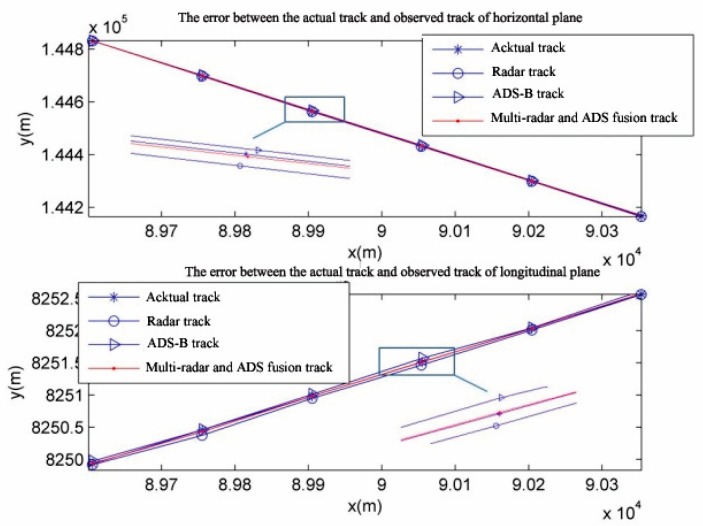
Observed track with weighted fusion.

**Figure 9 sensors-19-04975-f009:**
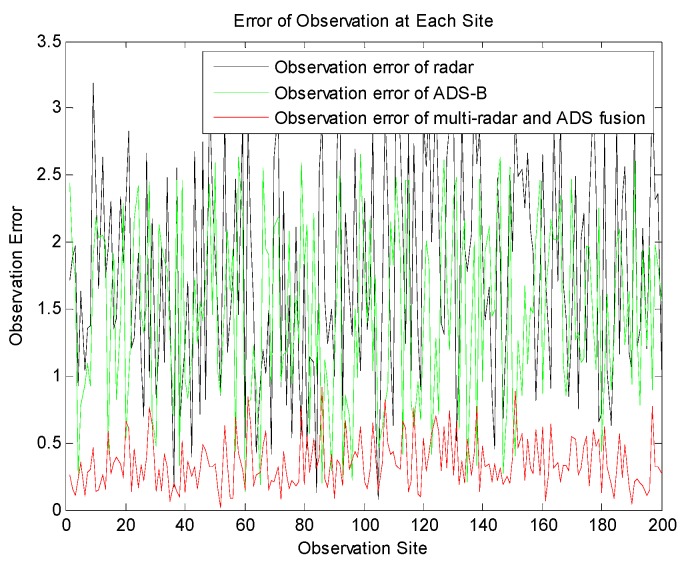
Comparison of the error of observation at each site.

**Figure 10 sensors-19-04975-f010:**
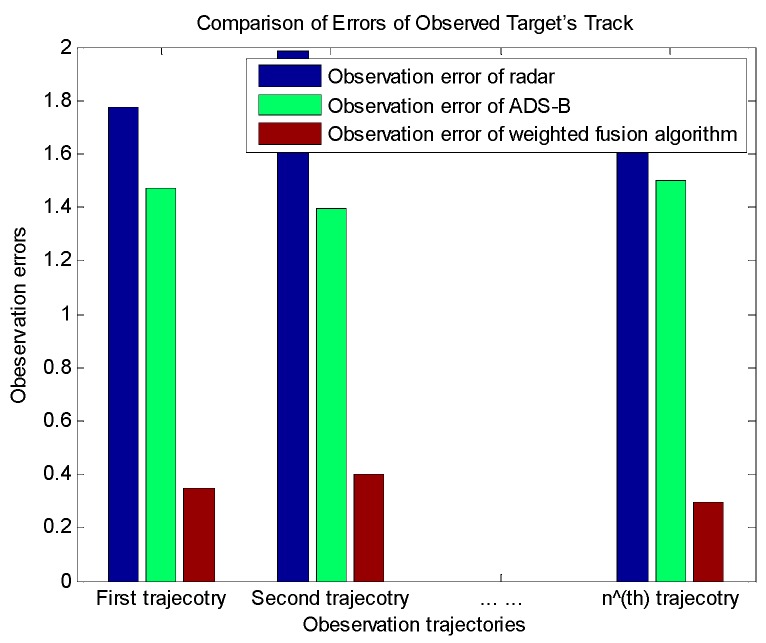
Comparison of errors of the observed target’s track.

**Figure 11 sensors-19-04975-f011:**
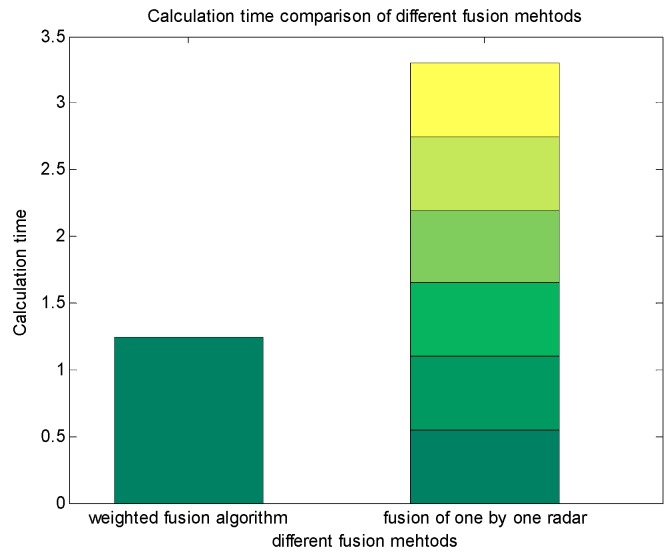
Comparison of calculation time with weighted fusion algorithm and general approach.

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
