# Peer review of "Research into a Multi-Variate Surveillance Data Fusion Processing Algorithm"

_sensors, 2019, doi:10.3390/s19224975_

Round 1

Reviewer 1 Report

The authors present a study on data fusion, essentially with data achieved from an airplane. The main defect of this paper is that the reader is immediately immersed in the technical. The author should introduce at least two points: 

1) what is the problem they concentrate on, what are the applications, what are the practical perspectives of their work.

2) What is new, improved, tested, in their work with respect to the literature on the topic dealt with.

Finally, the authors should all the times put a bibliographic reference when quoting works from othe authors. E.g. when they write: Gning uses Bernoulli PF.....and slightly after "Zhang uses BPF....

Reviewer 2 Report

The paper entitled "Research on Multi-variate Surveillance Data Fusion
Processing Algorithm" is an interesting approach to integrate different forms of surveillance in ATM in terms of data fusion.

Structure of paper: Authors must improve quality of presentation. Paper seems to be chaotic. Some chapters end with figure without any comment. Abstract does not show either reason or the backgroud of taking up this problem. The authors underestimate the complex problem of surveillance in ATM and do not show any literature on it. Introduction does not represent the severity of the problem. It is also hard to read figures 2,7,8 which makes paper less readible. The quality of English and style must be significantly improved. 

But, my main concern focuses on the idea of choosing only ADS and radars together as a system. Surveillance in ATM is much more complex. There are many types of radars giving different information. There are different operating modes of transponder (at least S or C) which makes problem not too easy. Multilateration MLAT or Wide Area Multilateration have been completely omitted, and are a separate group of completely different sensors. Test verification based on a sample of 15 aircraft positioning routes can hardly be called reliable. Authors do not mention the interoperability of presented method.

The summary is laconic, there are no conclusions.

Round 2

Reviewer 2 Report

I accept the paper in present form.